# The Power of Self-Compassion and PERMA+4: A Dual-Path Model for Employee Flourishing

**DOI:** 10.3390/bs15121620

**Published:** 2025-11-25

**Authors:** Scott I. Donaldson, Margaret L. Kern, Martin Suchta, Michelle McQuaid, Stewart I. Donaldson

**Affiliations:** 1Division of General Internal Medicine, Rutgers Robert Wood Johnson Medical School, New Brunswick, NJ 08901, USA; 2Centre for Wellbeing Science, University of Melbourne, Parkville, VIC 3010, Australia; peggy.kern@unimelb.edu.au (M.L.K.);; 3The Good Foundry, Melbourne, VIC 3930, Australia; 4Department of Psychology, Claremont Graduate University, Claremont, CA 91711, USA

**Keywords:** PERMA+4, self-compassion, workplace well-being, mediation, latent class analysis

## Abstract

A growing body of research has linked PERMA+4 to employee well-being and performance, yet the role of self-compassion as a psychological mechanism remains unexplored. This study tested a dual-pathway model using mediation and latent profile analyses to examine how PERMA+4 and self-compassion jointly influence work-related outcomes. A sample of full-time employees (N = 576) completed an online survey assessing PERMA+4, self-compassion, and psychological functioning. Mediation analyses showed that self-compassion partially mediated the relationship between PERMA+4 and PsyCap (β = 0.10), JAWS-positive (β = 0.04), JAWS-negative (β = −0.08), job stress (β = −0.10), and turnover intentions (β = −0.05). Latent profile analysis identified two psychological profiles: Flourishers and Strugglers. Flourishers reported higher well-being across all outcomes. These findings offer support for a dual-pathway model of flourishing and provide practical implications for organizations seeking to improve employee well-being through integrated positive psychology interventions targeting both PERMA+4 and self-compassion.

## 1. Introduction

[31] ([31]) PERMA model, comprising Positive Emotion (i.e., experiencing joy), Engagement (i.e., becoming deeply absorbed in work or activities), Relationships (i.e., feeling socially connected and supported), Meaning (i.e., having purpose), and Accomplishment (i.e., striving for greatness), has emerged as a prominent framework for understanding the building blocks of well-being. In addition to the five building blocks of PERMA, [30] ([30]) encouraged scholars to consider additional building blocks that may enhance the comprehensiveness and practical utility of the framework. As such, [7] ([7]) proposed four additional components: mindset (i.e., maintaining a growth-oriented attitude), physical health (i.e., sustaining vitality), environment (i.e., creating supportive and functional surroundings), and economic security (i.e., feeling financially stable and secure). This expanded model, referred to as PERMA+4, has been applied primarily in workplace contexts, where a growing body of empirical research has found it to be a robust and consistent predictor of employee well-being and positive functioning ([4]). Together, PERMA+4 can be considered a holistic framework for understanding employee flourishing in organizational settings.

Research has suggested that the addition of other positive psychological constructs may amplify the effects of employee well-being on performance ([1]; [3]; [8]; [19]; [24]; [32]; [34]). For example, [10] ([10]) found that the associations between PERMA+4 and work-related outcomes were significantly stronger when employees also possessed psychological safety in their work environment, suggesting that certain psychological resources can enhance the benefits of PERMA+4. These findings point to the potential for synergistic effects between well-being constructs. However, relatively little is known about which psychological characteristics may strengthen the effects of PERMA+4 on work outcomes.

Self-compassion is a theoretically grounded positive psychological resource that enables individuals to respond to personal failures and workplace stressors with kindness, mindfulness, and a sense of shared humanity ([15]; [20], [21]; [23]; [22]). From a theoretical perspective, self-compassion aligns with models of emotional regulation and coping, providing a buffer against self-criticism and emotional exhaustion that often accompany high-stress occupational settings ([13]). In recent years, a growing body of research has examined the role of self-compassion in the workplace ([6]). Findings suggest that self-compassion is associated with lower burnout, greater job satisfaction, improved psychological functioning, and enhanced emotional resilience among employees ([6]; [26]).

While prior research has examined PERMA+4 and self-compassion independently, few studies to our knowledge have empirically tested self-compassion as a mediating mechanism linking PERMA+4 to employee well-being outcomes. As such, the mediating role of self-compassion in the relationship between PERMA+4 and key psychological and work-related outcomes remains underexplored. Moreover, few studies have identified distinct psychological profiles based on combinations of PERMA+4 and related variables or examined how these profiles correspond to employee positive functioning. To address these gaps, the present study will examine the effects of PERMA+4 on employee well-being outcomes via self-compassion. Findings can highlight to what extent self-compassion links PERMA+4 and employee well-being, inform workplace well-being interventions, and advance the integration of self-compassion within applied positive psychology frameworks.

## 2. Materials and Methods

### 2.1. Procedures and Participants

As part of an ongoing longitudinal study involving 1200 participants, a panel agency helped recruit 600 individuals at an 18-month follow-up. All original participants were contacted again and invited to fill out an additional online survey. The inclusion criteria were the same as in the initial study: participants needed to be full-time employees, part of a team of at least two people, and have a direct supervisor (please cf. [10], for additional details on study design). Informed consent was collected before participation. Each participant received $3.00 for completing the follow-up survey. Out of the original participants, 24 did not finish the survey, leading to an analytical sample of 576 individuals. The research materials and procedures for the follow-up study were approved by the Institutional Review Board at Claremont Graduate University.

Table 1 presents the sociodemographic breakdown of this sample. A majority of participants identified as male (62.3%, n = 359), followed by female (37.2%, n = 214), with one participant identifying as another gender (0.17%) and two declining to respond (0.35%). The average participant age was 38.9 years (SD = 10.1). Educational attainment was highest at the bachelor’s level for 51.0% of the sample (n = 294), followed by master’s degrees (23.1%, n = 133), associate degrees (11.5%, n = 66), other forms of education such as high school diplomas or technical training (11.5%, n = 66), and doctoral degrees (3.0%, n = 17). In terms of racial and ethnic identity, most participants identified as White/Caucasian (71.2%, n = 410), with additional representation from Asian (9.2%, n = 53), Hispanic/Latino (7.8%, n = 45), Black/African American (7.6%, n = 44), and multiracial backgrounds (4.2%, n = 24). Participants reported employment across a variety of sectors, with the most common industries including software & IT services (17.4%, n = 100), military and other government-related sectors (14.2%, n = 82), and education (11.5%, n = 66). The full list of industries is provided in Table 1. Reported annual income ranged from less than $25,000 (14.9%, n = 86) to more than $150,000 (5.2%, n = 30), with the most frequent income bracket falling between $25,000 and $49,999 (33.9%, n = 195). Additional income distributions included $50,000–$74,999 (19.8%, n = 114), $75,000–$99,999 (15.8%, n = 91), $100,000–$150,000 (10.1%, n = 58), and two participants who declined to report income (0.30%).

### 2.2. Measures

PERMA+4: The PERMA+4 short scale was used to measure the nine building blocks of well-being ([9]). Participants were asked to reflect on their workplace experiences over the past two weeks and rate their level of agreement with nine statements representing each building block of well-being. Response options ranged from 1 = strongly disagree to 7 = strongly agree. Example items included “I felt physically healthy” and “I had a positive mindset at work.” Prior studies have supported the reliability and validity of the PERMA+4 short scale among diverse international samples ([10], [11]).

Self-compassion: Self-compassion was assessed using the 12-item Self-Compassion Scale–Short Form (SCS-SF; [25]). The scale includes two items from each of six subdomains: self-kindness, self-judgment, common humanity, isolation, mindfulness, and over-identification. Participants were asked to indicate how frequently they typically acted in each manner using a 5-point Likert scale (1 = almost never to 5 = almost always). The SCS-SF has demonstrated strong reliability and validity across diverse populations and languages, with a near-perfect correlation with the original 26-item long form ([25]).

Psychological capital: Psychological capital (PsyCap) was measured using an eight-item version of the Psychological Capital Questionnaire (PCQ; [18]), which captures its four dimensions: hope, self-efficacy, resilience, and optimism. Each construct was assessed with two items, and participants rated their agreement on a 7-point Likert scale ranging from 1 (strongly disagree) to 7 (strongly agree). The PsyCap measure has been widely validated in organizational research and is consistently linked to favorable work outcomes, including job satisfaction and performance ([18]).

Job-related affective well-being: Participants’ emotional experiences at work were assessed using the 20-item short form of the Job-Related Affective Well-Being Scale (JAWS; [33]). The instrument captures both positive (e.g., calm, enthusiastic) and negative (e.g., anxious, frustrated) emotional states, with items categorized by both valence and arousal level (i.e., high vs. low energy emotions). Respondents indicated how frequently they experienced each emotion in relation to their job over the past two weeks, using a 5-point scale ranging from 1 (never) to 5 (extremely often). The JAWS has been validated across a wide range of industries and cultural settings, demonstrating strong reliability and construct validity in occupational research ([33]).

Job stress: Perceived job stress was assessed using a five-item scale ([17]). This measure captures emotional and cognitive responses to occupational strain. Participants responded to items such as “My job frequently makes me feel frustrated or angry” and “I often experience tension or stress while at work,” indicating their agreement on a 5-point Likert scale (1 = strongly disagree, 5 = strongly agree). Prior research has demonstrated that this measure possesses strong internal consistency and construct validity across a variety of work environments ([17]).

Turnover intentions: Employees’ intentions to leave their current roles were measured using the six-item Turnover Intention Scale (TIS-6; [2]). The scale includes items that assess both the frequency and likelihood of considering job departure, such as “How often do you dream about getting another job that will better suit your personal needs?” Response formats varied slightly across items, with some rated from 1 (never) to 5 (always), and others from 1 (highly unlikely) to 5 (highly likely). The TIS-6 has shown strong internal reliability and robust predictive validity in organizational studies ([2]).

### 2.3. Analytic Strategy

Descriptive statistics, skewness and kurtosis values, and internal consistency estimates (Cronbach’s α) were calculated for all study variables ([5]). To examine the associations between PERMA+4, self-compassion, and employee well-being outcomes, multiple linear regression models were estimated. Model estimates were examined for statistical significance (i.e., *p* < 0.05), effect size, and variance explained (*R*^2^). To assess whether self-compassion mediated the relationship between PERMA+4 and outcome variables, a series of path models were specified. Indirect effects were calculated using bias-corrected bootstrap samples. Standardized coefficients and model fit indices were reported. Mediation was considered significant if the bootstrapped confidence interval for the indirect effect did not include zero. Next, a person-centered approach was employed to identify latent psychological profiles based on standardized PERMA+4 and self-compassion scores. Latent profile class analysis was conducted using Gaussian mixture modeling. Model selection was guided by Bayesian Information Criterion (BIC), and the best-fitting solution was used to assign participants to classes. Descriptive comparisons of outcome variables across latent classes were conducted to explore how well-being profiles related to outcomes of interest. All analyses were conducted in *R* (version 2023.12.1+402) using the lavaan, mclust, and psych packages ([27]; [28]; [29]).

## 3. Results

Descriptive statistics showed that skewness and kurtosis values for all study variables were within acceptable ranges, suggesting that the data were approximately normally distributed. Table 2 shows the means, standard deviations, correlations, and internal consistencies of PERMA+4, self-compassion, and well-being outcome measures. Internal consistency values ranged from acceptable to excellent across all measures (Cronbach’s α = 0.83–0.94). There was a strong, positive association between PERMA+4 and PsyCap (r = 0.73, *p* < 0.05), JAWS-positive (r = 0.68, *p* < 0.05), and self-compassion (r = 0.49, *p* < 0.05), and a strong negative association with JAWS-negative (r = −0.67, *p* < 0.05), job stress (r = −0.57, *p* < 0.05), and turnover intentions (r = −0.67, *p* < 0.05). Self-compassion showed similar patterns of association with all outcome variables. These results suggest that both PERMA+4 and self-compassion are strong predictors of work-related well-being outcomes.

As shown in Table 3, PERMA+4 demonstrated a strong total effect on PsyCap (β = 0.73, 95% CI [0.67, 0.79]), with a significant indirect effect through self-compassion (β = 0.10, 95% CI [0.07, 0.13]), indicating partial mediation. For JAWS-positive, the total effect was also strong (β = 0.59, 95% CI [0.53, 0.64]), with a smaller indirect effect (β = 0.04, 95% CI [0.01, 0.08]). For JAWS-negative, job stress, and turnover intentions, PERMA+4 was a significant negative predictor, with total effects ranging from β = −0.45 to −0.55. Indirect effects through self-compassion were also significant (βs = −0.05 to −0.10), indicating partial mediation for each outcome. *R*^2^ values ranged from 0.471 to 0.674 (Table 3). These findings suggest that PERMA+4 is a robust predictor of work-related well-being outcomes, and that self-compassion may help explain the strength of these associations.

Figure 1 displays the results from an LPA of PERMA+4 and self-compassion, identifying two distinct psychological profiles: Flourishers and Strugglers. The two-class solution was selected as the best-fitting model based on the Bayesian Information Criterion (BIC = −3101.81). Flourishers (n = 330) were characterized by high scores on both PERMA+4 and self-compassion, whereas Strugglers (n = 246) reported lower scores on both constructs. As shown in Figure 1, Flourishers reported substantially more favorable work outcomes than Strugglers. They exhibited higher levels of PsyCap (M = 5.73, SD = 0.70) and JAWS-positive (M = 3.37, SD = 0.68), and lower levels of job stress (M = 2.27, SD = 0.77), JAWS-negative (M = 1.88, SD = 0.60), and turnover intentions (M = 2.33, SD = 0.70). In summary, these findings reinforce the value of PERMA+4 and self-compassion in distinguishing profiles of employee positive functioning.

## 4. Discussion

This study showed that self-compassion partially mediated the relationship between PERMA+4 and employee well-being outcomes, providing initial support for a dual-pathway framework in which PERMA+4 influences employee well-being both directly and indirectly through self-compassion.

These findings extend prior research on the PERMA+4 framework by identifying self-compassion as a meaningful psychological mechanism through which PERMA+4 influences employee well-being. While PERMA+4 has been shown to predict a range of positive and negative workplace outcomes ([4]; [7]), the present study adds to this literature by demonstrating that self-compassion partially mediates these relationships, providing new support for the idea that PERMA+4 may exert its effects in the workplace, in part, by enhancing emotional resilience and self-regulatory capacity. Furthermore, self-compassion may not only mediate the effects of PERMA+4, but also moderate or amplify them over time. This aligns with past research ([6]), which highlighted the promise of self-compassion as a workplace resource but called for empirical studies that clarify its role within broader well-being frameworks ([16]).

Similar to prior research ([14]), this study identified two distinct psychological profiles—Flourishers and Strugglers—based on employees’ levels of PERMA+4 and self-compassion. Flourishers, who reported high levels of both resources, scored significantly higher across all indicators of employee well-being. In contrast, Strugglers exhibited comparatively low levels of PERMA+4 and self-compassion and reported poorer well-being outcomes. These findings suggest that a person-centered perspective may be necessary when studying workplace well-being. Rather than assuming uniform effects of individual variables, latent profile analyses can demonstrate how combinations of psychological resources cluster within individuals and relate to employee well-being outcomes. These psychological profiles may help organizations identify subgroups of employees with distinct needs and opportunities for interventions.

From a practical perspective, these results suggest that interventions targeting both PERMA+4 and self-compassion may be effective in facilitating employee flourishing across diverse work settings. Identifying and supporting employees with low levels of both resources may be a critical first step in building a resilient and high-functioning workforce. Tailored interventions that address both foundational well-being (e.g., PERMA+4) and self-directed emotional competencies (e.g., self-compassion) may offer organizations a comprehensive strategy for improving retention, morale, and psychological health.

### Limitations

Data were self-reported, which may have introduced common method variance and self-report bias ([12]). While the study was cross-sectional in design, the mediation models tested directional hypotheses. This precluded the research team from establishing temporal precedence. The sample was drawn from an online panel agency, which may limit generalizability to other occupational contexts or cultural settings.

## 5. Conclusions

This study examined a dual-pathway model of employee flourishing, showing that self-compassion partially mediated the effects of PERMA+4 on employee well-being outcomes. These findings underscore the importance of addressing both structural and emotional aspects of flourishing at work. The results may be useful for organizational leaders, HR professionals, and workplace well-being practitioners aiming to support employee well-being and performance. Future research should employ longitudinal or experimental designs to confirm causal mechanisms, explore potential reciprocal and moderating effects of self-compassion, and test integrated interventions that build both PERMA+4 and self-compassion in diverse occupational settings.

## Figures and Tables

**Figure 1 behavsci-15-01620-f001:**
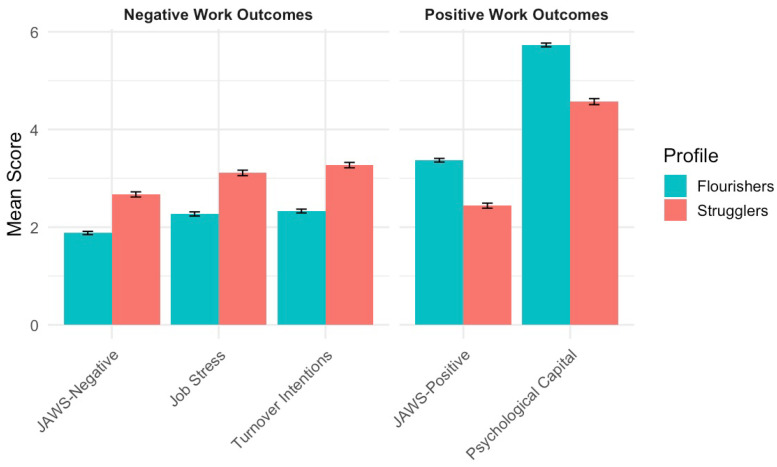
Mean scores on work-related outcomes by latent profile classification (Flourishers vs. Strugglers). *Note.* JAWS = job-related affective well-being.

**Table 1 behavsci-15-01620-t001:** Participants’ sociodemographic characteristics (N = 576).

Demographic Variable	n or M	% or SD
Gender		
Male	359	62.3
Female	214	37.2
Decline to state	2	0.35
Other (e.g., transgender)	1	0.17
Age	38.9	10.1
Education		
Bachelor	294	51.0
Master	133	23.1
Associate	66	11.5
Other (e.g., high school diploma)	66	11.5
Doctorate	17	3.0
Race/ethnicity		
Caucasian	410	71.2
Asian	53	9.2
Hispanic	45	7.8
Black	44	7.6
Multiracial	24	4.2
Industry		
Software & IT services	100	17.4
Other (e.g., military, hospitality)	82	14.2
Education	66	11.5
Retail, wholesale, & distribution	66	11.5
Healthcare	64	11.1
Government	54	9.4
Banking & financial services	52	9.0
Manufacturing	51	8.9
Media & entertainment	16	2.8
Food & beverage	14	2.4
Non-profit	11	1.9
Income		
Decline to state	2	0.30
Less than $25,000	86	14.9
$25,000–$49,999	195	33.9
$50,000–$74,999	114	19.8
$75,000–$99,999	91	15.8
$100,000–$150,000	58	10.1
More than $150,000	30	5.2

**Table 2 behavsci-15-01620-t002:** Means, standard deviations, correlations, and internal consistencies of PERMA+4, self-compassion, and well-being outcome measures.

Measure	M (SD)	1	2	3	4	5	6	7
1. PERMA+4	5.21 (1.01)	**0.89**						
2. Self-compassion	3.20 (0.77)	0.49	**0.89**					
3. Psychological capital	5.24 (1.01)	0.73	0.51	**0.89**				
4. JAWS-positive	2.97 (0.87)	0.68	0.41	0.59	**0.89**			
5. JAWS-negative	2.22 (0.80)	−0.67	−0.49	−0.60	−0.58	**0.94**		
6. Job stress	2.63 (0.91)	−0.57	−0.45	−0.56	−0.63	0.79	**0.87**	
7. Turnover intentions	2.73 (0.90)	−0.67	−0.42	−0.53	−0.71	0.69	0.67	**0.83**

*Note.* N = 576; JAWS = job-related affective well-being; bold values in the diagonals represent Cronbach’s alpha values; all correlation coefficients were statistically significant at *p* < 0.05.

**Table 3 behavsci-15-01620-t003:** Effects of PERMA+4 on well-being outcomes, mediated by self-compassion.

Outcome Variable	Direct Effect	95% CI (Direct)	Indirect Effect	95% CI (Indirect)	Total Effect	95% CI (Total)	R^2^
PsyCap	0.63	[0.56, 0.70]	0.10	[0.07, 0.13]	0.73	[0.67, 0.79]	0.562
JAWS-positive	0.55	[0.48, 0.61]	0.04	[0.00, 0.08]	0.59	[0.53, 0.64]	0.471
JAWS-negative	−0.45	[−0.51, −0.39]	−0.08	[−0.11, −0.05]	−0.54	[−0.58, −0.49]	0.670
Job stress	−0.44	[−0.51, −0.36]	−0.10	[−0.14, −0.06]	−0.53	[−0.59, −0.47]	0.587
Turnover intentions	−0.55	[−0.62, −0.49]	−0.05	[−0.09, −0.02]	−0.60	[−0.66, −0.55]	0.674

*Note.* All coefficients are standardized estimates from structural equation models with 5000 bootstrap samples. Confidence intervals represent percentile bootstrap 95% CIs. PsyCap = psychological capital; JAWS = Job-Related Affective Well-Being Scale.

## Data Availability

Data will be made available upon request.

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
