# Peer review of "The Power of Self-Compassion and PERMA+4: A Dual-Path Model for Employee Flourishing"

_behavsci, 2025, doi:10.3390/bs15121620_

Round 1
Reviewer 1 Report
Comments and Suggestions for Authors
This paper makes a clear and valuable contribution to the growing body of literature on workplace well-being and positive organizational psychology. It integrates two well-established frameworks, PERMA+4 and self-compassion, into a unified, empirically tested dual-pathway model of employee flourishing. The manuscript is theoretically sound, methodologically rigorous, and practically relevant for both scholars and practitioners.
The study is the first to empirically test self-compassion as a mediating mechanism in the relationship between PERMA+4 and employee well-being outcomes. This represents a conceptual bridge between foundational well-being structures (PERMA+4) and emotional-regulatory resources (self-compassion), enriching both frameworks. By proposing and testing a model where flourishing arises from both direct and indirect (self-compassion mediated) paths, the paper advances the theoretical understanding of how well-being components interact dynamically rather than functioning as parallel predictors. The inclusion of self-compassion highlights the intrapersonal mechanisms that may underlie previously observed organizational-level effects of PERMA+4, offering a more complete multilevel perspective. The paper situates itself within “Positive Organizational Psychology 2.0,” aligning well with current shifts toward systemic and integrative models of well-being (e.g., combining cognitive, affective, and behavioral elements).
The combination of mediation analysis (testing mechanism) and latent profile analysis (LPA) (identifying well-being subgroups) is particularly innovative. This dual approach provides both variable-centred and person-centered insights, a methodological complement rarely applied in this field. The study uses validated and psychometrically strong instruments, all with excellent internal consistency, increasing confidence in measurement validity. The dataset is demographically varied across sectors, education levels, and income groups, increasing the ecological validity and generalizability of the findings. The paper provides full standardized coefficients, bootstrap confidence intervals, and R² values, and describes model selection for LPA (via BIC), demonstrating high statistical transparency.
The identification of two employee well-being profiles (Flourishers and Strugglers) has diagnostic and intervention potential for HR and organizational development practice. The results point toward dual-focused interventions that combine PERMA+4 development (structural and cognitive dimensions) with self-compassion training (emotional-regulatory dimensions). These implications are communicated clearly and align with the paper’s theoretical premises.
Suggestion: consider potential reciprocal effects: given the bidirectional nature of well-being constructs, it would be valuable to acknowledge that self-compassion might not only mediate but also amplify or moderate the PERMA+4–outcome relationship over time.
In sum, this paper offers an important theoretical and empirical advance in understanding the psychological mechanisms of workplace flourishing, and it has a strong impact on the organizational well-being literature.
Author Response
REVIEWER 1
This paper makes a clear and valuable contribution to the growing body of literature on workplace well-being and positive organizational psychology. It integrates two well-established frameworks, PERMA+4 and self-compassion, into a unified, empirically tested dual-pathway model of employee flourishing. The manuscript is theoretically sound, methodologically rigorous, and practically relevant for both scholars and practitioners.
Response: Thank you for reviewing our manuscript.
The study is the first to empirically test self-compassion as a mediating mechanism in the relationship between PERMA+4 and employee well-being outcomes. This represents a conceptual bridge between foundational well-being structures (PERMA+4) and emotional-regulatory resources (self-compassion), enriching both frameworks. By proposing and testing a model where flourishing arises from both direct and indirect (self-compassion-mediated) paths, the paper advances the theoretical understanding of how well-being components interact dynamically rather than functioning as parallel predictors. The inclusion of self-compassion highlights the intrapersonal mechanisms that may underlie previously observed organizational-level effects of PERMA+4, offering a more complete multilevel perspective. The paper situates itself within “Positive Organizational Psychology 2.0,” aligning well with current shifts toward systemic and integrative models of well-being (e.g., combining cognitive, affective, and behavioral elements).
Response: Thank you for reviewing our manuscript.
The combination of mediation analysis (testing mechanism) and latent profile analysis (LPA) (identifying well-being subgroups) is particularly innovative. This dual approach provides both variable-centred and person-centered insights, a methodological complement rarely applied in this field. The study uses validated and psychometrically strong instruments, all with excellent internal consistency, increasing confidence in measurement validity. The dataset is demographically varied across sectors, education levels, and income groups, increasing the ecological validity and generalizability of the findings. The paper provides full standardized coefficients, bootstrap confidence intervals, and R² values, and describes model selection for LPA (via BIC), demonstrating high statistical transparency. The identification of two employee well-being profiles (Flourishers and Strugglers) has diagnostic and intervention potential for HR and organizational development practice. The results point toward dual-focused interventions that combine PERMA+4 development (structural and cognitive dimensions) with self-compassion training (emotional-regulatory dimensions). These implications are communicated clearly and align with the paper’s theoretical premises.
Response: Thank you for reviewing our manuscript.
Suggestion: consider potential reciprocal effects: given the bidirectional nature of well-being constructs, it would be valuable to acknowledge that self-compassion might not only mediate but also amplify or moderate the PERMA+4–outcome relationship over time.
Response: We agree that the relationship between PERMA+4, self-compassion, and employee outcomes is likely dynamic and reciprocal in nature. We now acknowledge in the Discussion section that self-compassion may also act as a potential amplifier of PERMA+4’s effects over time. We have added language to the manuscript noting that future longitudinal and experimental research is needed to test these alternative pathways and to explore the possibility of reciprocal and interactive effects among well-being resources.
In sum, this paper offers an important theoretical and empirical advance in understanding the psychological mechanisms of workplace flourishing, and it has a strong impact on the organizational well-being literature.
Response: We appreciate your review.
Reviewer 2 Report
Comments and Suggestions for Authors
Abstract Section Feedback:
Appropriate and relevant- the objective, methodology, significance, findings, implications and limitations are well brought out. From the onset-first sentence, the essence of the paper is clear, capturing its contribution to Psychology literature, beyond focused analysis on employee well-being and performance by introducing self-compassion to evaluate the influence on work-related performance. The findings, methodology, objectives are specific and clearly presented.
Introduction Section Feedback:
In-depth and appropriate- The PERMA Framework is deeply explained, though, other elements of Self-Compassion, linkages with well-being and outcomes should have preceded the introduction of the model. The problem statement and theoretical grounding in Psychology is clear. “To date, no studies have examined how PERMA+4 and self-compassion jointly influence employee well-being.”- this statement should be sufficiently justified.
Conceptually, clarify what has been done before, in relation to studying self-compassion.
Methodology Feedback:
The use of quantitative methodology is appropriate. The mediation and analytical tests are sufficient. Since it is a path-way model, more figurative illustrations could boost the readers’ understanding. The study participants and other ethical declarations have been made satisfactorily.
Results and Discussion:
The study is relevant; the output of the data analysis is clearly presented in the tables and discussions tied back into the PERMA+4 Model and Self-compassion studies which has enhanced consistency and coherence.
Overall Feedback: Thank you for this paper. It is concisely written, apt and timely.
Citation and References: Appropriate, recent and sufficient.
Author Response
REVIEWER 2
Abstract Section Feedback:
Appropriate and relevant- the objective, methodology, significance, findings, implications and limitations are well brought out. From the onset-first sentence, the essence of the paper is clear, capturing its contribution to Psychology literature, beyond focused analysis on employee well-being and performance by introducing self-compassion to evaluate the influence on work-related performance. The findings, methodology, objectives are specific and clearly presented.
Response: Thank you for reviewing our work.
Introduction Section Feedback:
In-depth and appropriate- The PERMA Framework is deeply explained, though, other elements of Self-Compassion, linkages with well-being and outcomes should have preceded the introduction of the model. The problem statement and theoretical grounding in Psychology is clear. “To date, no studies have examined how PERMA+4 and self-compassion jointly influence employee well-being.”- this statement should be sufficiently justified. Conceptually, clarify what has been done before, in relation to studying self-compassion.
Response: We have clarified that although prior research has explored PERMA+4 and self-compassion independently, no studies to our knowledge have empirically tested self-compassion as a mediating mechanism linking PERMA+4 to employee well-being outcomes, justifying the contribution of our study.
Methodology Feedback:
The use of quantitative methodology is appropriate. The mediation and analytical tests are sufficient. Since it is a path-way model, more figurative illustrations could boost the readers’ understanding. The study participants and other ethical declarations have been made satisfactorily.
Response: We have opted not to include additional figures because we believe Table 3 provides the most concise and interpretable format for comparing patterns of effects across the five outcome variables. Given the consistent dual-path structure across outcomes, separate diagrams would be redundant and potentially reduce clarity. That said, if the inclusion of visual figures is deemed necessary, we would be happy to produce them.
Results and Discussion:
The study is relevant; the output of the data analysis is clearly presented in the tables and discussions tied back into the PERMA+4 Model and Self-compassion studies which has enhanced consistency and coherence.
Response: Thank you for reviewing our work.
Overall Feedback: Thank you for this paper. It is concisely written, apt and timely.
Citation and References: Appropriate, recent and sufficient.
Response: Thank you for reviewing our work.
Reviewer 3 Report
Comments and Suggestions for Authors
Dear authors.
The topic of the article is highly interesting, and the use of the PERMA+4 approach adds rigour to the article.
I identified the following positives in the article:
- Abstract: The structure of the abstract is clear.
- Introduction: An appropriate combination of original and current sources is used.
- Methodology: This section is appropriately structured and contains clear information explaining the procedure, the use of the PERMA+4 method and details about the respondents.
- Results: The findings are clearly presented in the form of tables and figures. This section contains exact values.
I identified shortcomings in several parts:
- There is no separate "theoretical background" section. If the paper is included in the "article" category, I think that it should have the terms defined in a separate section according to the usual structure. I recommend splitting the current introduction - leaving general information in it and including specific explanations of terms in the newly created theoretical background section. The new chapter should contain the definition of the main terms from a theoretical perspective - definitions and descriptions. For example, defining what is: Employee Flourishing; Self-Compassion; defining the PERMA models (what they focus on, what can be achieved thanks to their application). (In case the authors decide not to create a new separate theoretical section, it is necessary to finalise the introduction and add new definitions and sources.)
- In the results section, I recommend adding a verbal explanation, a description of the results – the interpretation of the results is key for a better understanding of the conclusions.
- The article does not contain information about the future potential direction of research. I recommend expanding section 4.1. to “Limitations and future directions”.
- Discussion contains text that acts as a summary of the article. I recommend revising the discussion and adding references that will show a comparison with the findings of other authors. Some parts of the discussion can be moved directly to the conclusions (since this section is processed as only one paragraph).
- The number of references is low. 20 references are not enough for a publication in the “article” category. I recommend adding references mainly to the newly created theoretical part and discussion.
- Text formatting and editing:
- There should not be two headings in the text directly following each other. After the heading “2. Materials and Methods”, I recommend adding a short text for the introduction to this chapter.
- The chapter should not end with a table or figure. I recommend adding a brief summary: under Table 1 (before heading 2.2. Measures) and also after Figure 1.
Reviewer
Author Response
REVIEWER 3
The topic of the article is highly interesting, and the use of the PERMA+4 approach adds rigour to the article.
I identified the following positives in the article:
- Abstract: The structure of the abstract is clear.
- Introduction: An appropriate combination of original and current sources is used.
- Methodology: This section is appropriately structured and contains clear information explaining the procedure, the use of the PERMA+4 method and details about the respondents.
- Results: The findings are clearly presented in the form of tables and figures. This section contains exact values.
Response: Thank you for reviewing our work.
I identified shortcomings in several parts:
- There is no separate "theoretical background" section. If the paper is included in the "article" category, I think that it should have the terms defined in a separate section according to the usual structure. I recommend splitting the current introduction - leaving general information in it and including specific explanations of terms in the newly created theoretical background section. The new chapter should contain the definition of the main terms from a theoretical perspective - definitions and descriptions. For example, defining what is: Employee Flourishing; Self-Compassion; defining the PERMA models (what they focus on, what can be achieved thanks to their application). (In case the authors decide not to create a new separate theoretical section, it is necessary to finalise the introduction and add new definitions and sources.)
Response: Thank you for this thoughtful suggestion. We have opted to retain a unified Introduction section, consistent with the structure of similar empirical articles published in this journal and our prior work. In response to your feedback, however, we have carefully revised the Introduction to ensure that key theoretical constructs, such as PERMA+4, self-compassion, and employee flourishing, are clearly defined and appropriately sourced. This strengthened theoretical framing supports the rationale for the dual-pathway model tested in our study.
- In the results section, I recommend adding a verbal explanation, a description of the results – the interpretation of the results is key for a better understanding of the conclusions.
Response: Thank you for the suggestion. While we appreciate the request to include interpretive commentary in the Results section, we have opted to maintain a conventional structure in which statistical findings are presented without interpretation. Interpretation and implications are instead provided in the Discussion section.
- The article does not contain information about the future potential direction of research. I recommend expanding section 4.1. to “Limitations and future directions”.
Response: We agree that describing future research directions is important. These directions are provided in the Conclusion section, where we highlight the need for longitudinal and experimental studies to examine causal mechanisms and test integrated interventions that build both PERMA+4 and self-compassion.
- Discussion contains text that acts as a summary of the article. I recommend revising the discussion and adding references that will show a comparison with the findings of other authors. Some parts of the discussion can be moved directly to the conclusions (since this section is processed as only one paragraph).
Response: Thank you for the suggestion. We have made edits to the Discussion that incorporate references to prior research and clarify how our findings align with or extend existing literature.
- The number of references is low. 20 references are not enough for a publication in the “article” category. I recommend adding references mainly to the newly created theoretical part and discussion.
Response: Thank you for the suggestion. We have expanded the number of references throughout the manuscript, particularly in the Introduction and Discussion, to strengthen the theoretical foundation and situate our findings within the broader literature.
- Text formatting and editing:
- There should not be two headings in the text directly following each other. After the heading “2. Materials and Methods”, I recommend adding a short text for the introduction to this chapter.
Response: Thank you for the suggestion. We appreciate the formatting recommendation. However, we have opted to retain the current structure in line with the journal’s formatting template. If you deem this correction necessary, we will fix it.
- The chapter should not end with a table or figure. I recommend adding a brief summary: under Table 1 (before heading 2.2. Measures) and also after Figure 1.
Response: Thank you for the suggestion. We appreciate the formatting recommendation. However, we have opted to retain the current structure in line with the journal’s formatting template.
Round 2
Reviewer 3 Report
Comments and Suggestions for Authors
Dear authors.
I appreciate the refinement of the Introduction section and the addition of future directions.
However, the total number of resources increased by only 1. Although you added several references to the text (highlighted in yellow), most of them have already been used elsewhere in the article.
No additional information on the interpretation of the results was added to the results section.
Given the current state of the article, I recommend:
- Expand the number of references by at least 10 (so that the total number of references in the article is at least 30).
- Refine the interpretation of the findings in the results (at least in three places in this section, considering the tables and graph used).
Reviewer
Author Response
Given the current state of the article, I recommend:
- Expand the number of references by at least 10 (so that the total number of references in the article is at least 30).
Response: Thank you for the suggestion. We have expanded the reference list to include additional relevant citations across the Introduction and Discussion sections, bringing the total number of references above 30.
- Refine the interpretation of the findings in the results (at least in three places in this section, considering the tables and graph used).
Response: Thank you for the helpful suggestion. In response, we added brief interpretive statements to three locations within the Results section, each aligned with a major analysis and corresponding table or figure. These additions highlight the practical significance of the findings and clarify the predictive value of PERMA+4 and self-compassion for workplace outcomes.